# Designing a Laboratory Bioassay for Evaluating the Efficacy of Antifouling Paints on *Amphibalanus amphitrite* Using a Flow-Through System

**Ryuji Kojima** [1,*], **Seiji Kobayashi** [2], **Kiyotaka Matsumura** [3], **Cyril Glenn Perez Satuito** [4], **Yasuyuki Seki** [5], **Hirotomo Ando** [1] **and Ichiro Katsuyama** [2]

1   Department of Marine Environment and Engine System, National Maritime Research Institute, Mitaka, Tokyo 181-0004, Japan; ando@nmri.go.jp
2   Department of Environmental Risk Consulting, Japan NUS Co., Ltd., Shinjuku, Tokyo 166-0023, Japan; kobayasi@janus.co.jp (S.K.); katuyama@janus.co.jp (I.K.)
3   School of Marine Bioscience, Kitasato University, Sagamihara, Kanagawa 252-0373, Japan; matsumurasipc@gmail.com
4   Graduate School of Fisheries and Environmental Sciences, Nagasaki University, Nagasaki 852-8131, Japan; satuito@nagasaki-u.ac.jp
5   Hiroshima R&D Centre, Chugoku Marine Paints, Ltd., Otake, Hiroshima 739-0652, Japan; yasuyuki_seki@cmp.co.jp
*   Correspondence: kojima@nmri.go.jp; Tel.: +81-422-41-3769

**Abstract:** With the aim of establishing a protocol for evaluating the efficacy of antifouling paints on different organisms, a flow-through laboratory test using triangular boxes was developed for cyprids of the barnacle *Amphibalanus* (=*Balanus*) *amphitrite*. Six different formulations of antifouling paints were prepared in increasing content (0 to 40 wt.%) of $Cu_2O$, which is the most commonly used antifouling substance, and each formulation of paint was coated on one surface of each test plate. The test plates were aged for 45 days by rotating them at a speed of 10 knots inside a cylinder drum with continuously flowing seawater. The settlement behavior of 3-day-old cyprids released inside triangular boxes made from the test plates was observed. A decreasing number of juveniles settled on surfaces of test plates that were coated with paint containing more than 30 wt.% of $Cu_2O$. Results of the laboratory bioassays were consistent with those from the field experiments.

**Keywords:** antifouling efficacy; flow-through; triangular box; *Amphibalanus amphitrite*; cuprous oxide; dynamic aging; repellant activity; raft experiment; bioassay; biofouling of ships' hull

## 1. Introduction

A wide range of macrofoulers have been used as test organisms in antifouling bioassays conducted under controlled experimental conditions [1,2]. Barnacles are typical fouling organisms that attach to ships' hulls and submerged artificial structures. This biofouling consequently leads to increased fuel consumption and accidental introduction of non-indigenous species to another marine environment, possibly causing significant and harmful changes [3–12]. Controlling the attachment of barnacles is of great significance in the development of antifouling technology.

Internally fertilized eggs of barnacles' hatch when embryos develop in the mantle cavity of brooding adults, and they then start to drift in the ocean. During the planktonic larval stage, barnacles molt through six naupliar stages before they molt to cyprids that do not feed. Cyprids attach to a suitable substrate and metamorphose into juveniles to start a sessile life stage. This process of

attachment and metamorphosis is most important in the life cycle of barnacles. The inhibition of this process is the key to the development of antifouling technology on barnacles.

Many laboratory studies using cyprids have been conducted to elucidate the settlement mechanism of barnacles and to search for anti-foulants [13–19]. The barnacle *Amphibalanus amphitrite* (*A. amphitrite*) is widely distributed in the intertidal zones of the subtropical and temperate regions of the globe [16–18]. The establishment of its larval rearing method has made its cyprids available in the laboratory all year round [16]. They are also used worldwide as a model species in larval settlement studies of barnacles [20,21]. Most studies have been carried out under a static condition [20,22–31]. Nevertheless, antifouling agents are designed to be slowly released. This means that in an evaluation test for antifouling efficacy, the accumulation of the antifouling agent becomes a huge concern under a static condition. Therefore, the ideal assay condition should be a flow-through water system that does not allow the accumulation of antifouling agents in the test water. Re-circulating the test water inside the tank was proposed as a new settlement assay method to address problems encountered under a static water condition [18,32].

A new flow-through bioassay was reported by Pansch et al. [15] as a tool for rapid laboratory-based screening of candidate compounds for use in antifouling coatings. In a previous investigation, the authors designed a bioassay with a flow-through water system and successfully assessed the efficacy of antifouling paints using the mussel—*Mytilus galloprovincialis* (*M. galloprovincialis*) [33]. The flow-through bioassay system designed for *M. galloprovincialis* was unique in that it was compact and cost-effective, since bioassay vessels that were used were small and required a lesser amount of seawater during the test. In order to comprehensively assess the efficacy of antifouling paints, it is important to conduct laboratory bioassays on more than one test organism, since sensitivity to biocide may vary among fouling species. Therefore, the validation of laboratory bioassays for assessing the efficacy of antifouling paints will require broadening the objectivity of the laboratory experiment by including other fouling organisms. Hence, the authors designed a flow-through bioassay system for *A. amphitrite* that incorporated the features of our previous system.

In this study, the authors proposed and validated a newly developed laboratory bioassay for evaluating the efficacy of biocide-releasing antifouling paints with in a flow-through system using *A. amphitrite*. In order to conduct this study, test paints containing the antifoulant cuprous oxide ($Cu_2O$) were prepared in varying concentrations ranging from 0 to 40 wt.%. To simulate the actual condition of ship hulls, test plates coated with the test paints were initially cured dynamically and the inhibition effects of the paints was evaluated using cyprid settlement as the index. A comparison of results from laboratory bioassays using barnacle larvae and from field experiments conducted on rafts is also discussed.

## 2. Materials and Methods

### 2.1. AF Paints and Test Plates

Six types of AF paints with vinyl copolymer coatings and containing 0, 5, 10, 20, 30, and 40 wt.% of $Cu_2O$ were prepared, as shown in Table 1. Polyvinyl chloride (PVC) plates used in laboratory experiments were 50 mm × 50 mm × 2 mm in size (Kasai Sangyo Co., Ltd., Osaka, Japan). For the experimental groups, the test plates had the same size as their control counterparts, and were coated on one side with the test paint, as specified in the Performance Standards for Protection Coatings (PSPC) [34]. Surface treatment, measurement of $Cu_2O$ concentration and the dynamic aging process of the plates were conducted according to the previous paper [33].

**Table 1.** Composition of the test paints used [33]. Adapted with permission from [33]. Copyright 2016 PLOS.

| Composition/Paint Name | A-0 | A-1 | A-2 | A-3 | A-4 | A-5 |
|---|---|---|---|---|---|---|
| Cuprous oxide | 0 | 5 | 10 | 20 | 30 | 40 |
| Xylene | 23 | 23.6 | 24 | 25 | 26 | 27 |
| Methylisobutylketone | 5 | 5 | 5 | 5 | 5 | 5 |
| Base polymer | 9 | 8.7 | 8.5 | 8 | 7.5 | 7 |
| Rosin | 9 | 8.7 | 8.5 | 8 | 7.5 | 7 |
| Barium sulphate | 50 | 45 | 40 | 30 | 20 | 10 |
| Anhydrous ferric oxide | 1 | 1 | 1 | 1 | 1 | 1 |
| Oxidized polyethylene wax | 1 | 1 | 1 | 1 | 1 | 1 |
| Amide wax | 2 | 2 | 2 | 2 | 2 | 2 |

Values in the table indicate mass %.

*2.2. Laboratory Bioassay*

2.2.1. Test Organism

Adults of *A.* (=*Balanus*) *amphitrite* were collected from Lake Hamana (Shizuoka, Japan: 34°41′ N, 137°35′ E). These were reared in the laboratory according to the method described in previous reports [18,19,23,35–39]. Adults of *A. amphitrite* were transferred into a tank filled with seawater adjusted to a temperature of 25 °C and were fed nauplii of the brine shrimp, *Artemia salina*. The seawater used was cartridge (1 μm) filtered and seawater diluted with purified water (Milli-Q, Merck Millipore, Burlington, MA, USA) to adjust its salinity to 28 ± 0.5. The seawater was renewed daily to maintain the water quality in the rearing tank. To induce the release of nauplius larvae, adults were taken out of the tank and kept dry inside the incubator at 25 °C for at least 6 h. Nauplius larvae were released when adults were submerged back in the seawater adjusted to 25 °C. Newly hatched nauplius larvae exhibited positive phototactic behaviour and gathered around a light source. They were collected using a pipette and cultured according to the method described in a previous paper [19]. Briefly, clean two-liter glass beakers were used for culturing larvae of barnacles. For larval cultures of nauplii to the cyprid stage, 0.22 μm (Nylon filter membrane, Merck, Kenilworth, NJ, USA) of filtered seawater with the salinity adjusted to 28 using purified water was used. Streptomycin sulfate (FUJIFILM Wako Pure Chemical Corporation, Osaka, Japan) and Penicillin G sodium salt (FUJIFILM Wako Pure Chemical Corporation) were added into the culture water to final concentrations of 30 and 20 μg/mL, respectively. Cyprids were obtained after 5 days when cultured following the conditions presented in Table 2.

**Table 2.** Culture condition for nauplius larvae.

| Condition | Remarks |
|---|---|
| Seawater | 0.22 μm filtered seawater (Nylon filter membrane, Merck) with the salinity adjusted to 28 using purified water |
| Density of larvae | 2 to 3 larvae per 1 mL |
| Diet and density | The diatom *Chaetoceros gracilis* (200,000 to 400,000 cells/mL), other diatoms can also be used, such as *Skeletonema costatum* (1,000,000 to 2,000,000 cells/mL) |
| Antibiotics | Streptomycin sulfate (30 μg/mL), Penicillin G sodium salt (20 μg/mL) |
| Water temperature | 25 ± 1 °C |
| Light | The light intensity is approximately 3000 lux., and the photoperiod was 12 h. light, 12 h. dark period |
| Aeration | Approximately 20 mL/min. |

Cyprids were collected in a beaker filled with 0.22 µm of filtered seawater and stored in a refrigerator at a temperature of 4 to 6 °C for about three days, before their use in bioassays. The condition of cyprids used in the bioassays for each culture batch was checked by observing mortality and settlement rates. That is, 10 individuals of 3-day-old cyprids were placed in each well of a 12-well polystyrene plate, with each well filled with 4 mL of the filtered seawater, and the number of juveniles that settled and dead cyprids were counted under a dissecting microscope after 48 h [40–42].

### 2.2.2. Evaluating the Inhibition Effects of Test Paints Using Cyprid Settlement as Index

The bioassay system consisted of the seawater storage tank (volume capacity of 20 L), peristaltic pump, polypropylene bioassay tank (diameter × depth = 80 mm × 55 mm, volume capacity = ca. 275), and reservoir tank (height × width × depth = 1500 mm × 900 mm × 450 mm) for waste seawater. The seawater storage tank and reservoir tank were placed outside the incubator. The schematic diagram of the system is shown in Figure 1. The size of the bioassay tank used was large enough to completely submerge the test plates. The bioassay tank also had a siphon tube made of glass for drainage. The bioassay tank was placed inside an incubator equipped with a temperature controller to keep temperature within 25 ± 1 °C. The set-up was illuminated for 12 h. each day with a light intensity of 3000 lux. During the bioassays, 0.22 µm filtered seawater was continuously charged into the bioassay tanks at a flow rate of 7 mL/min. Three rounds of bioassays were performed. Throughout the investigation, a control group with the control plates was always set and assessed simultaneously with the other test plates.

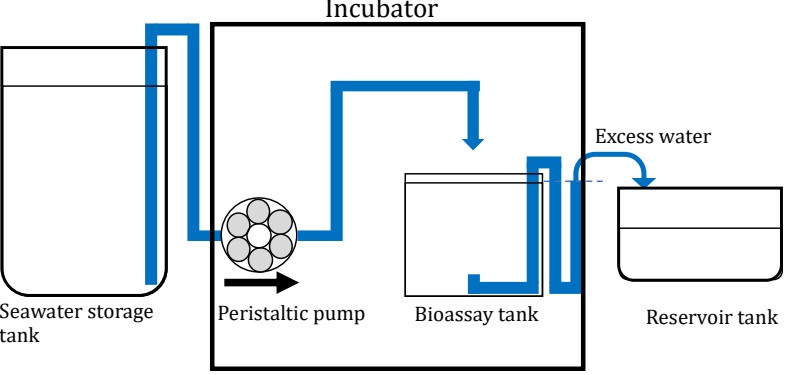

**Figure 1.** Schematic diagram of the bioassay system.

The bioassay was conducted using a triangular box, designed to enhance the settlement of cyprids by enclosing them in a small space within the three test plates. The box had one test/control plate and two white acrylic plates, assembled as shown in Figure 2. The triangular box was assembled so that the test or the control surface was positioned facing inward the triangular box. We found that larvae hardly settled on the surface of white acrylic plate (data not shown). Therefore, we used the white acrylic plates to promote cyprid settlement on the control and test surface.

Prior to the experiment with antifouling paints, a suitable control plate was determined using the present bioassay. Initially, the bioassay described above was used to determine a suitable control plate for inhibition experiments. Materials tested for the selection of a control plate were the white colored polystyrene (PS) plate, the grey colored PVC plate and the black colored PVC plate (Kasai Sangyo Co., Ltd., Osaka, Japan). Various treatments were also applied on the surface of the control plate by blasting with polishing agents, after the suitable material of the control plate was selected, and cyprid settlement on treated surfaces was investigated.

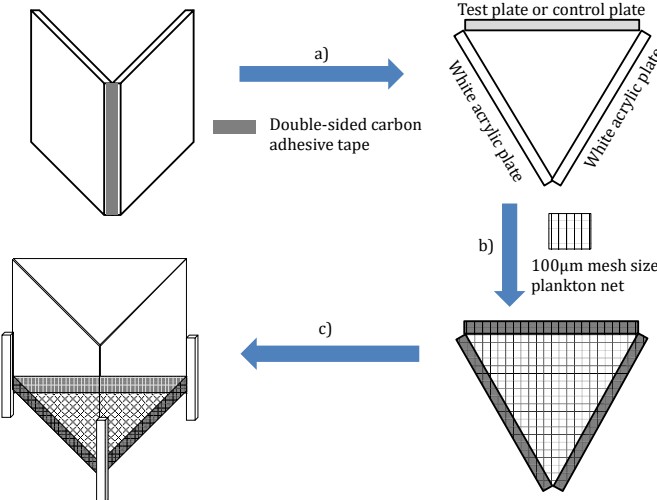

**Figure 2.** Assembling of the triangular box for the bioassay. (**a**) One test plate (or control plate) and two white acrylic plates were assembled using a carbon adhesive tape to form a triangular box; (**b**) the bottom-side of the triangular box was covered with a 100-μm mesh sized plankton net using a carbon adhesive tape; (**c**) plastic rods were attached to the triangular box to serve as legs for elevation (length of the rods from the bottom-side of the box: >10 mm).

The surfaces of the test and control plates were kept wet during the assembly of the triangular box. The bottom of the triangular box was covered with plankton net (mesh size NXX13, pore size: 100 μm). The test or control plate, two acrylic plates, and the plankton net were tightly adhered together using double-sided carbon tape (Nisshin EM Co., Ltd., Tokyo, Japan) as this material does not affect the swimming behavior of cyprids. Three plastic rods were attached to the triangular box to serve as legs, in order to elevate the bottom-side (net side) of the box by at least 10 mm from the bottom of the bioassay tank (Figure 3).

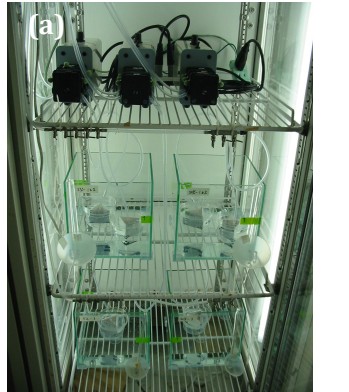
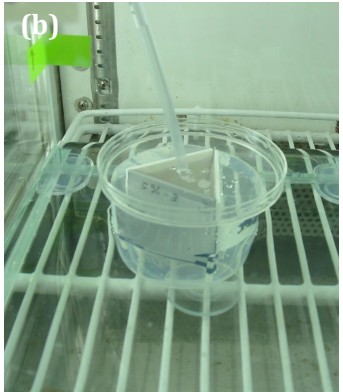

**Figure 3.** Photographs showing the bioassay system. (**a**) A photograph of peristaltic pumps and the triangular box inside the bioassay tanks placed in the incubator; (**b**) a photograph of the triangular box used in the bioassay.

The triangular boxes were assembled one day prior to the bioassay and then immersed overnight in the test seawater at room temperature. During this time, the seawater was stirred using a glass rod to prevent its stagnation near the triangular box. The triangular box was later taken out from the tank and immersed in 1 L of seawater for five minutes prior to the bioassay. The triangular box was then positioned inside the bioassay tank and the tank placed inside the incubator. The seawater was allowed to flow inside the bioassay tank from the topside of the triangular box. The flow rate was adjusted to prevent overflowing of seawater from the triangular box, as shown in Figure 4.

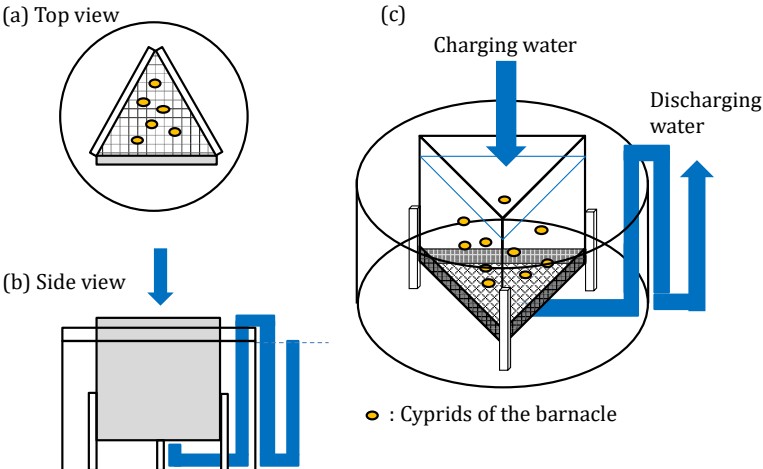

**Figure 4.** The triangular box deployed in the bioassay tank. (**a**) Top view of a triangular box containing cyprids deployed in the bioassay tank; (**b**) side view of a triangular box inside the bioassay tank; and (**c**) arrows indicating the direction of the flow of the test seawater being charged inside the triangular box and discharged from the bioassay tank.

One hundred cyprids were released inside the triangular box. Initial water temperature, salinity (CM-14P, DKK-TOA Corp., Tokyo, Japan), and pH (HM-30P, TKK-TOA Corp., Tokyo, Japan) of the test seawater in the bioassay tank were measured and monitored during the bioassay. Unattached cyprids and dead individuals inside the triangular box were immediately collected after measuring the water quality parameters at the end of the experiment. The triangular box was immediately disassembled after retrieving the cyprids and dead individuals. The number of settled juveniles, cyprids and dead individuals on each of the three surfaces of the triangular box and on the net were counted under a stereo microscope.

### 2.2.3. Verification of the Validity of the Bioassay

The number of settled juveniles on a multi-well plate, cyprids, and dead individuals were counted. The settlement ratio for verifying the validity of the bioassay ($R_v$) was calculated using Equation (1):

$$R_v = \frac{a}{a+b+c} \times 100 \tag{1}$$

where $a$ indicates the number of juveniles settled on the surface of the multi-well plate; $b$, the number of cyprids (not settled); $c$, the number of dead individuals; $R_v$, the settlement ratio for verifying the validity of the bioassay (%).

The number of juveniles that settled on the test plate and the other surfaces (acrylic plates, edges of the test plate and net), the cyprids and the dead individuals were counted. The settlement ratio for test plate ($R_t$) was also calculated using Equation (2):

$$R_t = \frac{S_t}{S_t + a + b + c} \times 100 \tag{2}$$

where $S_t$ indicates the number of settled juveniles on the surface of the test plate; $a$, the number of settled juveniles on the other surfaces; $b$, the number of cyprids (not settled); $c$, the number of dead individuals; $R_t$, the settlement ratio for test plate (%).

The settlement ratio for the control plate ($R_c$) was also calculated using Equation (3):

$$R_c = \frac{S_c}{S_c + a + b + c} \times 100 \tag{3}$$

where $S_c$ indicates the number of juveniles that settled on the surface of the control plate; $a$, the number of juveniles that settled on the other surfaces; $b$, the number of cyprids (not settled); $c$, the number of dead individuals; $R_c$, the settlement ratio for the control plate (%).

The average values of the settlement ratio in all the experimental and control rounds were calculated using Equations (4) and (5), respectively:

$$A_t = \frac{\sum_{j=1}^{j}(R_{t1}^j + R_{t2}^j + \cdots R_{tn}^j)}{\sum_{j=1}^{j} n^j} \tag{4}$$

where $j$ indicates the run number; $R_{tn}^j$, the settlement ratio of the $n$-th test plate on the $j$-th run (%); $n^j$, the number of test plates on the $j$-th run; $A_t$, the average value of the settlement ratio in the experimental round (%).

$$A_c = \frac{\sum_{j=1}^{j}\left(R_{c1}^j + R_{c2}^j + \cdots R_{cn}^j\right)}{\sum_{j=1}^{j} n^j} \tag{5}$$

where $j$ indicates the run number; $R_{cn}^j$, the settlement ratio of the $n$-th control plate on the $j$-th run (%); $n^j$, the number of control plates on the $j$-th run; $A_c$, the average value of the settlement ratio in the control round (%).

Finally, the relative settlement ratio of cyprids ($R$) was calculated using Equation (6):

$$R = \frac{A_t}{A_c} \tag{6}$$

### 2.3. Statistical Analysis

Statistical analysis, including one-way analysis of variance (ANOVA), nonparametric tests, and Holm-Sidak's multiple comparison test ($p < 0.05$) in the settlement assay, calculation of correlation coefficients, and variance analysis were performed using GraphPad Prism version 7.0 d for Mac OSX (GraphPad Software).

## 3. Results

### 3.1. Assessment of the Efficacy of Antifouling Paint on Cyprid Settlement Ratio

#### 3.1.1. Parameters of Water Quality of the Test Water

The water temperature, salinity, and pH of the test water in the three laboratory experiments were controlled at $24.4 \pm 1.2\ °C$, $27.4 \pm 0.9$, and $8.3 \pm 0.1$, respectively. The concentrations of $Cu_2O$ in the test water of the control groups ranged from 2.2 to 2.5 µg/L after 48 h. Whereas, the concentrations of $Cu_2O$ in the test water of the experimental groups were 2.3 µg/L (A-0), 2.9 µg/L (A-1), 13.3 µg/L (A-2), 4.7 µg/L (A-3), and 12.9 µg/L (A-4) after 48 h. The concentrations of $Cu_2O$ in the test water of the A-5 groups ranged from 20.1 to 26.0 µg/L in the three experiments.

#### 3.1.2. The Activity of Cyprids Used in the Bioassay

Settlement ratios ranged from 80% to 93%, and mortality was 0%. Therefore, cyprids used in the settlement assays were healthy and bioassays were verified as valid.

#### 3.1.3. Relative Settlement Ratios ($R$) of Cyprids on Antifouling Paints

In the bioassay for the selection of a suitable material for the control plate, settlement ratio after the 48-h immersion period was calculated using Equation 5. Three replicates were conducted for each type of plate. Averages of the settlement ratios were 50.1% (SD = 16.5) for the black colored PVC, 28.8 % (SD = 11.9) for the grey colored PVC, and 25.6% (SD = 17.5) for the white colored PS plates, respectively.

The mortality of cyprids was less than 0.4% in each group. In another experiment, the surface of the black colored PVC was blasted with F-40, F-36 and F-20 polishing agents [43,44] respectively, and settlement ratios were also investigated after the 48-h immersion period. Four replicates were prepared for each blasted plate. Averages of the settlement ratios for F-20, F-36, and F-40 blasted plates were 53.2% (SD = 14.7), 24.9% (SD = 5.4), and 25.5% (SD = 9.7), respectively. Furthermore, the settlement ratio for the F-20 blasted black PVC was investigated at different times for validation, with four replicate plates tested at each time. Averages of the settlement ratios were 53.3% (SD = 14.7), 56.0% (SD = 10.7), and 44.9 % (SD = 14.5) for the three times. Time-related difference in settlement ratios was not recognized by variance analysis ($F = 1.0358$, $df = 3$, $p = 0.4181$). As a result, the black color PVC plate blasted with F-20 was used as a control plate in this bioassay.

Experiments with antifouling paints were conducted between October, November, and December of 2014, and in February of 2015. Each experiment was replicated three times; using a total of nine test plates for each experimental group and three test plates for the control group. To assess the inhibition effect of the test paints, statistical analysis was conducted by comparing results from the experimental groups with their respective controls. Results of the mortality and settlement bioassay are shown in Figures 5 and 6, respectively. Average mortality of cyprids in the control group was less than 1%, while mortalities in the 0, 5, 10, 20, and 30 wt.% $Cu_2O$ experimental groups were less than 5%, and that in the 40 wt.% of $Cu_2O$ group was less than 10%.

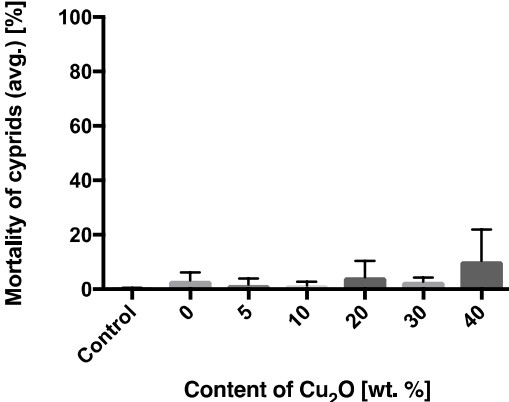

**Figure 5.** Average mortalities of cyprids in the control and experimental groups. Error bars indicate SDs.

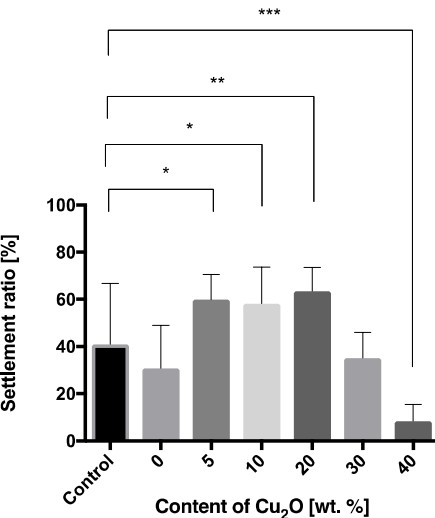

**Figure 6.** Settlement ratios in the control and experimental groups. Error bars indicate SDs. Pairwise comparisons of results between each experimental group and the control were conducted to establish significance of the difference (* $p < 0.05$, ** $p < 0.01$, and *** $p < 0.001$), where the symbols *, ** and *** corresponds to significant, very significant and extremely significant, respectively.

The settlement ratio in the control ($A_c$) was 40%, and settlement ratios in the 5 wt.% of $Cu_2O$, 10 wt.% of $Cu_2O$, and 20 wt.% of $Cu_2O$ experimental groups ($A_t$) were higher than that of the control. The values of the settlement ratio in the 5 wt.% and 10 wt.% of $Cu_2O$ experimental groups were significantly different from $A_c$ ($p < 0.05$). In the 20 wt.% of $Cu_2O$ group, the difference between $A_t$ and $A_c$ was very significant ($p < 0.01$), but there was no significant difference between $A_c$ and $A_t$ of the 30 wt.% of $Cu_2O$ group. Concentrations below 30 wt.% of $Cu_2O$ promoted settlement, whereas an inhibition of settlement was clearly observed at 40 wt.% of $Cu_2O$, where the difference in settlement ratio as compared to the control was extremely significant ($p < 0.0001$). To evaluate the settlement inhibition effect of the test paints, data were normalized by calculating the relative settlement ratios of cyprids ($R$) in the experimental groups with respect to their respective controls.

The $R$ values of the paints containing different concentrations of $Cu_2O$ are shown in Figure 7. $R$ values showed a non-linear relationship with the $Cu_2O$ content in the paint ($r^2 = 0.3403$, non-linear curve fitting, second-order polynomial (quadratic equation)), where $R$ increased with increasing $Cu_2O$ content of the paint, at up to 20 wt.%, but decreased thereafter. The results also showed that 50% settlement inhibition ($EC_{50}$) was obtained at approximately 38.0 wt.% of $Cu_2O$ through interpolation of the fitting curve. Moreover, the $R$ value at 40 wt.% of $Cu_2O$ was less than 0.5 and the relative settlement ratio in this group obviously differed from that in the other experimental groups. Therefore, $R$ values of 0.5 and lower can be considered as inhibition settlement ratio.

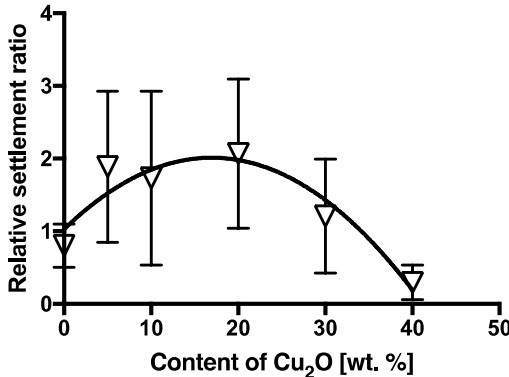

**Figure 7.** Relative settlement ratios on paints containing different concentrations of $Cu_2O$. Error bars on the closed triangles indicate SDs of the relative settlement ratios of cyprids.

## 4. Discussion

### 4.1. The Concept of the Flow-Through Bioassay Designed for Barnacles in the Laboratory

Biocides have been primarily screened in-vitro for antifouling activity and toxicity using multi-well plates [2]. Even experiments investigating the tolerance of nauplius larvae to copper stress have been designed in a still water condition [17,45]. It is evident that literature on the evaluation of the efficacy of antifouling agents mostly described bioassay conducted in a still water condition.

Under such a condition, the leached biocides accumulate in the experimental system as the experimental period progresses. This increase in the concentration of biocides may affect the physiological condition of the test organisms. As a result, it is difficult to appropriately conduct an experimental bioassay. To address this difficult situation, the authors designed an experimental method with a flow-through system that renewed the test water continuously inside the experimental system. This system followed the same concept of the flow-through bioassay designed for the mussel *M. galloprovincialis* [33]. It is unique compared to other flow-through bioassay systems previously reported (e.g., [15]) due to the following features: (a) it has a low (ca. ~0.42 L/h) flow rate and is compact (vessel volume ca. 275 mL); (b) a constant density of cyprids inside the triangular vessel can be adjusted and ensured; and (c) enclosing cyprids inside a triangular vessel and using inert white acrylic plates enhanced cyprid settlement on the surface of test plates, depending on the efficacy of

the antifouling coating on the test plate. The barnacle *Amphibalanus amphitrite* was selected as the test organism because it is reported as one of the major macrofoulers [46,47]. However, no literature, except for this study, has ever reported on a barnacle settlement assay that is compact and that simulated actual conditions of the painted surface of ship hulls. Moreover, no study prior to this has introduced a barnacle settlement assay with a flow-through system in a small vessel, thereby using only a small volume of seawater.

The material chosen for the control plate was black PVC because it made observation of settled juveniles easier. The surface of the control plates was blasted with the polishing agent, F-20. This combination of material and surface treatment of the control plates resulted in a higher settlement ratio in the control group.

### 4.2. The Repellent Effect of $Cu_2O$ on Barnacles

In a toxicity study conducted under a still water condition, the mortality of barnacles from $CuSO_4 \cdot 5H_2O$, which was used as a positive control, was 10% at 1000 µg/L; and mortality was 50% at 3000 µg/L [25]. The repellent effect on barnacles was detected from 100 µg/L with 20% inhibition of settlement [25]. The $EC_{50}$ value was 300 µg/L, and 90% inhibition was at 1000 µg/L [25]. At the end of the experiment in this study, the $Cu_2O$ concentration of the test water was 24 µg/L in the group coated with the paint A-5 containing 40 wt.% of $Cu_2O$. This concentration was almost 1/10 to 1/40 times lower than the value in the abovementioned still water bioassay [25]. Mortality of cyprids in the paint A-5 group was 10% after 48 h.

In this study, the $A_t$ values of 5, 10, and 20 wt.% of $Cu_2O$ groups were higher than that of the control. Hence, the concentration below 30 wt.% of $Cu_2O$ promoted cyprid settlement. The result indicates that tolerance to copper stress was below 20 wt.% of $Cu_2O$ content in the test paints. An indication of the relative tolerance of various organisms to toxic paints was reported previously [48]. *Amphybalanus amphitrite* was reported to be more tolerant to Cu than other foulers. The total number of adult barnacles attached to the toxic paint surface was less than 1% of those growing on the non-toxic control. *Amphybalanus amphitrite* comprised more than 90% of the total adult barnacle population on the toxic surface, although it comprised only 7.5% of the population attached to the control surface [48]. Regarding the toxic effect of copper on the larval development of barnacles, larvae in the advanced developmental stages may have developed better physiological mechanisms to regulate the uptake or increase the excretion of the toxicant [45]. As part of an antifouling investigation, the survival time and $O_2$ consumption of adult *A. amphitrite* exposed to different Cu concentrations were investigated [49]. The result showed that $O_2$ consumption rate of barnacles decreased during respiration, and data on survival time indicated that *A. amphitrite* was more tolerant to Cu than *B. tintinnabulum*. It was argued that tolerance to Cu was derived from metallothioneins [49]. These proteins were found to increase with exposure to metal concentrations, leading to the sequestration and detoxification of metals to some extent [49]. The concentration of copper in barnacles from uncontaminated sites was reported [50,51], and it demonstrated that the barnacle has the potential to accumulate high concentrations of copper and showed strong net accumulation of copper. In this situation, all incoming copper was accumulated for detoxification [52], and barnacles from Cu-contaminated sites had many type B (more heterogenous in shape and always containing sulphur in association with metals that include copper and zinc [53,54]) Cu-rich granules, probably resulting from lysosomal breakdown of metallothionein binding copper [51].

Another possible explanation of this phenomenon is the effect as an inducer. It has been demonstrated that chemical compounds such as epinephrine, phenylephrine, clonidine, KCl, $NH_4Cl$, and organic solvent induced mussel larval metamorphosis [55]. The authors explained that sub-toxic levels of these compounds could have triggered larval metamorphosis by physiologically "shocking" the larvae since dead larvae were observed in concentrations that induced metamorphosis [55]. In addition, the enhancement of the settlement of *Capitella* sp. I larvae by $H_2S$ had a sub-lethal effect [56]. It was also hypothesized that sub-lethal concentrations of $H_2S$ could trigger larval settlement and/or

metamorphosis by physiologically "shocking" the larvae [56]. In this report, the authors explained that the stimulus for settlement was a chemical cue associated with chemical substances to some extent. Copper was essential for crustaceans and mollusks, and excess amounts of chemical substances were toxic leading to the inhibition of the settlement of barnacles [52]. Such an increase in the settlement ratio of barnacles to $Cu_2O$ content that was between 5 and 20 wt.% occurred in the present study.

In order to validate the antifouling efficacy of the test paints, results of laboratory bioassays and field experiments were compared. The reason for this comparison is because data of biofouling obtained from raft and patch experiments are affected by geological and seasonal variations and assessing the performance of antifouling paints in these experiments takes time [57,58]. Details of the field experiments were reported by the authors in a previous paper [33]. The immersion period during the field experiments was 28 days, after which the degree of fouling on the test plates was evaluated. Figure 8 shows the relationship between *R* and ranks of fouling. Both the *R* and the ranks of fouling showed a similar decreasing tendency with increasing $Cu_2O$ content at more than 20 wt.% at St.1 and St.2 (St.1: $r^2 = 0.9759$, non-linear curve fitting, one-phase decay; St.2: $r^2 = 0.9135$, non-linear curve fitting, one-phase decay [33]). As a result, the proposed method using barnacles can sufficiently verify the repellant effect of test paints within 48 h. This method can also be used on other fouling species, such as algae, in future studies prior to the field experiment [59].

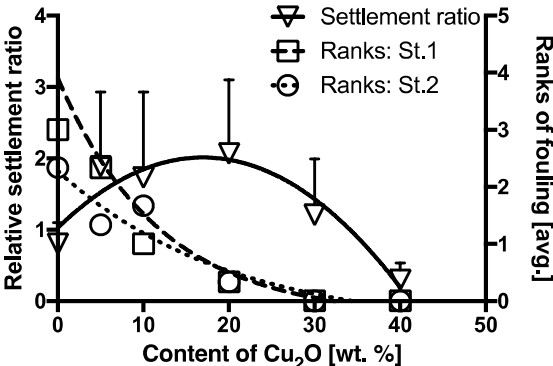

**Figure 8.** The relative settlement ratio of barnacle larvae and the degree of fouling in the field experiment on relation to the concentration of $Cu_2O$ on the test paints. The relative settlement ratios from bioassay experiments ($\triangledown$) and the degrees of fouling at St. 1 ($\square$) and St.2 ($\bigcirc$) cited from a previous paper [32] were plotted in relation to the concentration of $Cu_2O$ in the paints. Error bars on the open triangles indicate SDs.

## 5. Conclusions

The newly proposed method for evaluating the efficacy of biocide-releasing antifouling paints in a flow-through system using *A. amphitrite* was validated. A reproduceable and effective laboratory bioassay was established by evaluating the antifouling efficacy of test paints that were prepared with varying $Cu_2O$ contents. To simulate the actual condition of ship hulls, dynamic aging of test plates was conducted. A positive correlation between the $Cu_2O$ content and the repellant effect of the paint on barnacle larvae was observed at concentrations of more than 30 wt.%. Comparison of the results between laboratory bioassays using barnacle larvae and of field experiments revealed a highly consistent relationship between the two. Our results support the use of barnacles in evaluating the efficacy of antifouling paints because this bioassay does not test toxicity and is precise in that it imitates the exposed condition of paint on ship hulls, when the ship is in a stationary state after voyage. The novelty of this method lies in the aspect of assessing antifouling efficacy of paints by the evaluation of the behavior of barnacles inside the triangular box in a flow-through system, and so far, this is the only study that evaluates barnacle larval behavior to antifouling paints in situ in a flow-through system. This study also proved to be a significantly consistent method for assessing the effectiveness of present or future antifouling paints.

**Author Contributions:** Conceptualization, S.K., C.G.P.S., K.M., and I.K.; Data Curation, R.K., S.K., C.G.P.S., Y.S., and K.M.; Formal Analysis, R.K. and S.K.; Funding Acquisition, R.K. and H.A.; Investigation, R.K., S.K., C.G.P.S., Y.S., and K.M.; Methodology, R.K., S.K., C.G.P.S., K.M., and I.K.; Project Administration, R.K. and H.A.; Resources, H.A.; Supervision, H.A. and I.K.; Validation, S.K., C.G.P.S., and K.M.; Visualization, I.K.; Writing—Original Draft, R.K.; Writing—Review and Editing, S.K., C.G.P.S., and K.M.

**Funding:** This research was supported by the research project on "The study on management of biofouling (2012)" of the Japan Ship Technology Research Association (JSTRA) (project ID: 20011965347, continued project ID: 2000070441C99), which was funded by the Nippon Foundation (http://www.nippon-foundation.or.jp).

**Acknowledgments:** The authors would like to express their appreciation to researchers Hiroshi Masuda, Hirohisa Mieno, Kazuki Kouzai of Chugoku Marine Paints Co., Ltd., Mamoru Shimada of Nippon Paint Marine Coatings Co., Ltd., Eiichi Yoshikawa of ME consulting, and Tetsuya Senda of JSTRA for their useful comments and contribution to the study. Finally, the authors wish to acknowledge A.S. Clare of Newcastle University for his editing and constructive suggestions on the early version of this manuscript.

**Conflicts of Interest:** The authors have declared that no competing interests exist. The funders had no role in the design of the study; in the collection, analyses, or interpretation of data; in the writing of the manuscript, and in the decision to publish the results.

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
