# Peer review of "Designing a Laboratory Bioassay for Evaluating the Efficacy of Antifouling Paints on Amphibalanus amphitrite Using a Flow-Through System"

_coatings, doi:10.3390/coatings9020112_

Reviewer 1 Report

The authors reported an interesting approach in the field. However, and in order to provide enough data for others to accurately evaluate the work or ever to follow it, there is a lack of system description. Moreover, there is also a lack of comparison with similar systems in the field. Therefore, it is highly suggested to review other developed flow-through systems for the same purpose, emphasizing the main differences with the proposed one. For instance one of the most recent ones can be found in the paper:

Christian Pansch et al., A new flow-through bioassay for testing low-emission antifouling coatings”. Biofouling, Vol 3(8), 613-623, 2017

The whole system also lacks for a clear description:

How the whole system it placed in an incubator, which is a closed system, in order to maintain the water flow through it? It should include a pump system? For each experimental test are include two coated plates of the same formulation?

Why it is performed a prior panels immersion in the test seawater, and further quickly submersion in 1L of seawater? Is the test seawater reused for the bioassay test?

A photo of the system will help.

Other minor suggestions:

1.     Revise the English

2.     Revise values in %, for instance, it is in the abstract 0-40%, probably you mean 0-40 wt.%.

3.     Include a table with the prepared paint formulations, even if they were published before, it is not good for a reader to have to look for data included in the paper for analysis. In any case, those formulations should have been prepared for this work.

4.     Include the type of polymeric matrix of the coatings (polyurethane, acrylic, or other…)

5.     Avoid the use of similar sentence as found in reference [32].

6.     2.2.1 section, units for the salinity range

7.     Clarify in the text and figures the control for the cyprids culture and the panel control (0% of Cu2O) used for the relative settlement determination.

8.      Line 339-340: it is said that the system conditions imitate the field condition of a ship hull. This should be stated carefully, in fact, the conditions to which a ship hull is placed is the opposite, the coated surface moves through the seawater, being subjected to drag friction.

9.       Conditions (average temperature, salinity, pH, local coordinates) of the field test should be included allowing to compare it with the lab tests since the period was relatively short (28 days) and thus remained in one season condition. 

Reviewer 2 Report

This  laboratory bio assay flow-thorough system is well designed to mimic more closely the real conditions of  sea bio fouling. It ensures precise control over all operation conditions. In section 2. Materials and Methods the composition of the test AF paints is not presented. This raises the question if this assay is applicable for anti fouling efficiency evaluation of both contact killing and anti fouling agent releasing surfaces or for one of them. The conditions are quite different in both cases.The limitations of this laboratory bio assay should be noted obligatory. It is interesting if this assay will be relevant for evaluation the anti fouling efficiency against two or more macro fouling species presenting simultaneously in the flow -thorough system. 

Reviewer 3 Report

This is a nice study that should be published after attention is paid to scholarship, details in the methods and minor editing for standard English and word choice. 

Designing a laboratory bioassay for evaluating the 3 efficacy of antifouling paints on Balanus amphitrite 4 using a flow-through system.

Line comment:

19—Amphibalanus (= Balanus) Amphitrite-name change- official name is Amphibalanus

45—The original work on this topic is from the 80s on natural products and synthetic peptides.  In addition to the   1992 Rittschof et al reference, these should be cited.  The work can be found in the review article in Peptides by Rittschof and Cohen, 2004.

119—this is a triangular open ended box not really a prism. Prisms bend light.

Comment, why not make the triangular container all out of one type of material?  Justify this somewhere.

150—What were flow rates?  And how delivered single stream

The three plates in the container are not independent.  Please address this.

Once our barnacles set on the copper coatings do they stay alive?

Discussion

Pyfinch and Mott 1948 show toxicity to copper to barnacle larvae as<1/2 of what you report

And reports ISHIDA, S. (1936). Set. Pap. Inst. Phys. Chem. Res. Tokyo, 30, 195 found similar results.  Please put your findings in context.

Limits for copper release of antifouling coatings are<17 ug/square cm per day.  What were your release rates?

Author Response

Round  2

Reviewer 1 Report

Dear Authors,

Thank you for the improvements provided. I would just suggest reviewing some sentences, which are the same as used in your previous paper, for instance, lines 81-91 of page 3. 

Best success.

Author Response

We thank reviewer 1 for the re-careful reading of our revised manuscript and for giving us useful comments. 

Further, we would also grateful thank to your review in the period of between the last Christmas and the starting of this new year. 

I am going to correct and reflect the point in the whole part of the next revised manuscript in order not to use the similar sentence as found in our previous paper. 

In addition, English language editing is now being checked by a native English speaker.

Thank you very much.

Reviewer 2 Report

The reviesed manuscript is o.k. and can be published in this form.

Author Response

(The authors gave the same response as above.)

Reviewer 3 Report

Dear Authors,

This manuscript is NOT written in standard English and well over half of the world accepts the name change, now 12 years old, of Balanus to Amphibalanus. If you are unwilling acknowledge this, then this manuscript should not be published. 

Author Response

We thank reviewer 3 for the re-careful reading of our revised manuscript and for giving us useful comments. 

Further, we would also grateful thank to your review in the period of between the last Christmas and the starting of this new year. 

In the 1st round of the peer review, I made a big mistake without correction of the name of species due to my completely overlooking the point of the reviewer's comment.

I apologize my big mistake and thank you very much for teaching me the history of the species name.

I am going to correct and reflect the point in the whole part of the next revised manuscript.

In addition, English language editing is now being checked by a native English speaker.

Thank you very much.

Round  3

Reviewer 3 Report

Ok,

Now I can read this paper and I have just a few issues;

The methods are inadequate.  They need to be in sufficient detail to be repeated.

That means dimensions and sizes of containers, flow rates, and all the surface treatments including colors and how exactly the surfaces were blasted, 

shouldn't the control be coating without copper?  

the two white acylate sides have little settlement --less than the paint?  they are not inert 

Is your settlement ratio really just percentage settlement?

barnacles routinely settle more quickly and metamorphose before they die on toxic coatings

do you know they were alive after 48 hours?

Is your section on the comparison of lab and field intended to show that you don't get the same results?  That is what figure 8 shows. 

You device might be valuable for determining release of copper from the coatings.

Can you calculate release rates for copper in micrograms per square cm per day?

the US limit is 17 ug/square cm per day.   

Designing a laboratory bioassay for evaluating the

Line    comment

21.  add (=Balanus) after Amphibalanus and include in key words for continuity with older literature.

25.  Settlementbehavior

26.  made from 

27.  was observed

39.  Biofoulingleads to

43.  embryos develop  in the mantle cavity of brooding adults

93.  is  also discussed

118. A.(=Balanusamphitrite   do this once for continuity

147. sizes of sea water storage and reservoir tank so someone could use this set-up.

What was flow rate?

160. data not shown? Instead of unpublished?

164. control plates tested were white? clear? polystyrene, grey PVC, and Black PVC

Which material was selected?

191. put (Figure 4) at end of sentence instead of a whole sentence

258. in thesettlement assay

279. grey pvc

What color was the poly styrene?

283. need to put the different treatments in the methods section.

333. remove thebefore barnacles

350. explain why the white plates are “inert”

393.  How about reduction of bacterial film during the assay as an explanation for copper effects at intermediate concentrations?

409. this section belongs in results, not discussion. Then you can talk about it. 

Figure 8 the relation of the triangles to the open circles and squares is unclear. 

Field did not show increased settlement over control at middle concentrations.  This reviewer is not convinced. How is an inverted U shaped curve related to the two curves form the field. 

What are you trying to show?

Author Response

We thank reviewer for the careful reading of three times of our manuscript and for giving useful comments again. In response to the reviewer's comment, we have revised the manuscript. We look forward to hearing from you regarding our submission. We would be glad to respond to any further questions and comments that you may have. Our responses to the reviewer's comments are as follows:

1. Major comment

1-1. Dimensions and sizes of

(1) containers

Response 1-1: Storage tank has a volume capacity of 20 L. When the test seawater runs short during a bioassay, it is replaced with a new one. Regarding the bioassay tank, the used tank was a polypropylene cup (diameter: 80mm x depth: 55mm, capacity of volume: ca., 275ml). The reservoir tank used was a large container with dimensions of aapproximately 1500mm x 900mm x 450mm. The dimensions of the containers were included in our revised manuscript.

(2) flow rates:

Response 1-2: I already addressed this question at the 1st round of reviewers comment, and answered as follows:

“Response 4: The system which consisted of a test seawater tank, an incubator, a peristaltic pump and a reservoir tank for waste seawater was used for the bioassay. A syphon tube for drainage was connected to the bioassay tank to maintain the water flow-through at a flow rate of about 7 ml/min (less than 1L/h). Details regarding the flow rate was included in the revised manuscript.”

(3) all the surface treatments including colors and how exactly the surfaces were blasted

Response 1-3: Colors of control panels were prepared by a plate manufacturing company under high-quality control. The surface treatment was prepared by blasting with polishing agents specified in ISO or JIS standard, and confirmed by the measurement of surface roughness of the panels by a paint manufacture. The same procedure was used in our previous paper on mussels (Kojima et al. 2016), hence we reflected your comment in our revised manuscript.   

2. shouldn't the control be coating without copper?

Response 1-4: No, for the control panel we used black PVC blasted with polishing agent. The suitable control was determined as explained in the Materials and Methods, section 2.2.2, and the control panel selected consistently gave high settlement.

3. the two white acylate sides have little settlement --less than the paint?  they are not inert. Is your settlement ratio really just percentage settlement?

Response 1-5: Regarding the white plates, the plate had little settlement compared to the test panel coated with test paint within this study. Yes, the settlement ratio (R) was the percentage derived from the ratio between control and test panels. This was described in the Materials and Methods, section 2.2.3. 

4. barnacles routinely settle more quickly and metamorphose before they die on toxic coatings, do you know they were alive after 48 hours?

Response 1-6: Thank you very much for your comment. I already addressed this question at the 1st round. I answered the question as follows;

“Point 5: Once your barnacles set on the copper coatings do they stay alive?

Response 5: Most of attached juveniles on the test plates stayed alive after 48 hrs in the bioassay. You can see the average mortality of cyprids in Figure 3. Average mortality of cyprids in the control group was less than 1%, while mortalities in the 5 wt. %, 10 wt. %, 20 wt. % and 30 wt. % Cu2O experimental groups were less than 5%, and that in the 40 wt. % of Cu2O group was less than 10%.”

5. is your section on the comparison of lab and field intended to show that you don't get the same results?  That is what figure 8 shows. 

Response 1-7: In our previous report on mussels, the curves between laboratory bioassay and the field experiment were in good conformity with the degree of fouling and the same tendency in the due course of immersion period. Therefore, we also investigated the correlation and tendency between the laboratory bioassay and the field experiment in this paper. However, the inverted U-shaped curve derived from the laboratory did not match the curve from the field experiments. We discussed the inverted U-shaped curve from the viewpoint of “physiological shock” condition of barnacle when exposed to a certain concentration (sub-lethal concentration) of the biocide. This inverted U-curve phenomenon in the bioassay is a new finding in this study. However, the mechanism behind the phenomenon of enhanced settlement at certain concentrations (sub-lethal concentrations) still remains to be solved.

6. your device might be valuable for determining release of copper from the coatings. Can you calculate release rates for copper in micrograms per square cm per day? the US limit is 17 ug/square cm per day.

Response 1-8: Thank you very much for your useful comment. I already addressed the comment at the 1st round. Could you please refer to the previous comment as follows?

“Point 6: Pyfinch and Mott 1948 show toxicity to copper to barnacle larvae as<1/2 of what you report. And reports ISHIDA, S. (1936). Set. Pap. Inst. Phys. Chem. Res. Tokyo, 30, 195 found similar results.  Please put your findings in context. Limits for copper release of antifouling coatings are <17 ug/square cm per day.  What were your release rates?

Response 6: This bioassay is not a toxicity test of copper to barnacle. The paper you mentioned was conducted under a static water condition without a leaching agent from actual antifouling paints. Kitano et al, 2003, showed that the EC50 value was 300 μg/L, and 90% inhibition was at 1000 μg/L, and we just compared the value of the group coated with the paint A-5 containing 40 wt. % of Cu2O with the value from Kitano’s research for reference. The value we mentioned is not a toxicity data, if anything, but a data on settlement inhibition. This bioassay showed that these results support the use of barnacles in evaluating the efficacy of antifouling paints because this bioassay does not test toxicity and is precise in that it imitates the exposed condition of the paint on the ship’s hull at the stationary state after voyage.

Regarding the release rate of copper, we took water samples in the dynamic aging tank and simply measured before and after the dynamic aging of test panels. The dynamic aging of all panels was conducted at the same time. The figure A shows the copper-leaching rate at each content group after the dynamic aging. The concentration in dynamic aging tank was below 20 ppb after dynamic aging. The leaching rate of each panel was confirmed to be at a steady state after 45 days. At the 0 wt.% content group, the concentration was almost same as the level in natural seawater, and these results suggest that there was no copper absorption from the water in the dynamic aging tank. After the bioassay of 48 hrs., the leaching rate ranged from 13-22 ug/cm2/day in 40 wt.% content of cuprous oxide.”

Figure A. The copper-leaching rate at each content group after the dynamic aging.

2. Minor comment 

Point 1: Line 21   add (=Balanus) after Amphibalanus and include in key words for continuity with older literature. (OK)

Response 2-1: Thank you very much for your comments. However, I am very confused that reviewer 3 pointed Amphibalanus (= Balanus) Amphitrite-name change- official name is Amphibalanus” at the 1st and 2nd round. Anyway, I am going to follow the comment and add “(=Balanus)” after Amphibalanus at the pointed line, and reflected this in the revised manuscript.

Point 2: Line 25, Settlement behavior

Response 2-2: I reflected this in the revised manuscript.

Point 3: Line 26, made from 

Response 2-3: I reflected this in the revised manuscript.

Point 4: Line 27, was observed

Response 2-4: I reflected this in the revised manuscript.

Point 5: Line 39, Biofoulingleads to

Response 2-5: I reflected this in the revised manuscript.

Point 6: Line 43, embryos develop in the mantle cavity of brooding adults. 

Response 2-6: I reflected this in the revised manuscript.

Point 7: Line 93, is also discussed

Response 2-7: I reflected this in the revised manuscript.

Point 8: Line 118, A.(=Balanusamphitrite do this once for continuity

Response 2-8: I reflected this in the revised manuscript.

Point 9: Line 147, sizes of sea water storage and reservoir tank so someone could use this set-up. What was flow rate?

Response 2-9: Please refer to 1-1.

Point 10: Line 160 data not shown? Instead of unpublished?

Response 2-10: Yes, these data are not reported. However, we found that larvae hardly settled on the surface of white acrylic plate within this study. Therefore, we defined the white acrylic plate as an inert material within this study. We revised the term to “data not shown” in the revised manuscript.

Point 11: Line164, control plates tested were white? clear? polystyrene, grey PVC, and Black PVC. Which material was selected?

Response 2-11: Tested control plate in this study was PS, PVC with grey and black prior to the determination of suitable material for control plate of bioassay. After determination of suitable material for control plate, i.e., black PVC plate, the surface treatment of the control plate was investigated using polishing agent for blasting, specified in ISO or JIS.

Point 12: Line 191, put (Figure 4) at end of sentence instead of a whole sentence

Response 2-12: I reflected this in the revised manuscript.

Point 13: Line 258, in the settlement assay

Response 2-13: I reflected this in the revised manuscript.

Point 14: Line 279, grey pvc. What color was the poly styrene?

Response 2-14: The color of PS was white.

Point 15: Line 283, need to put the different treatments in the methods section.

Response 2-15: I reflected this in the revised manuscript as follows:

“Various treatments were also applied on the surface of the control plate by blasting with polishing agents, after the suitable material of the control plate was selected, and cyprid settlement on treated surfaces was investigated.”

Point 16: Line 333, remove the before barnacles

Response 2-16: I reflected this in the revised manuscript.

Point 17: Line 350, explain why the white plates are “inert”

Response 2-17: The white acrylic plates are inert surface compared to the control plate and of course, the test plates, for the settlement of cyprids in “this bioassay”.

This enable the triangular box to conduct the bioassay under the flow-through condition.

As explained in the paper, our preliminary experiment (data not shown) showed that larvae hardly settle on the surface of white acrylic plates, and this was mentioned in our manuscript. Less settlement of cyprids on whiteacrylic panels may be due to its smooth texture and the white color. It is generally perceived that cyprids settle more on black than white surfaces (Yasuda 1970).

Point 18: Line 393, How about reduction of bacterial film during the assay as an explanation for copper effects at intermediate concentrations?

Response 2-18: Thank you very much for your useful suggestion. We thought about the possibility of biofilm. However, the surface of test plates was cleaned and wiped off prior to the bioassay. Therefore, we considered that biofilm formation did not occurred during the bioassay at all.

Point 19: Line 409, this section belongs in results, not discussion. Then you can talk about it. 

Response 2-19: I reflected this on the final manuscript.

Point 20-1: Figure 8 the relation of the triangles to the open circles and squares is unclear.

Response 2-20: Please refer response 1-7, and response 2-21.

Point 20-2: Field did not show increased settlement over control at middle concentrations. This reviewer is not convinced. How is an inverted U-shaped curve related to the two curves form the field? What are you trying to show?

Response 2-21: In our previous report on mussels, the curves between laboratory bioassay and the field experiment were in good conformity with the degree of fouling and the same tendency in the due course of immersion period. Therefore, we also investigated the correlation and tendency between the laboratory bioassay and the field experiment in this paper. However, the inverted U-shaped curve derived from the laboratory did not match the curve from the field experiments. The authors tried to explain the inverted U-shaped curve from the viewpoint of “physiological shock” condition of barnacle when exposed to a certain concentration (sub-lethal concentration) of the biocide. This inverted U-curve phenomenon in the bioassay is a new finding in this study. However, the mechanism behind the phenomenon of enhanced settlement at certain concentrations (sub-lethal concentrations) still remains to be solved.